# Comparing Unsupervised Word Translation Methods Step by Step

**Mareike Hartmann**
Department of Computer Science
University of Copenhagen
Copenhagen, Denmark
hartmann@di.ku.dk

**Yova Kementchedjhieva**
Department of Computer Science
University of Copenhagen
Copenhagen, Denmark
yova@di.ku.dk

**Anders Søgaard**
Department of Computer Science
University of Copenhagen
Copenhagen, Denmark
soegaard@di.ku.dk

## Abstract

Cross-lingual word vector space alignment is the task of mapping the vocabularies of two languages into a shared semantic space, which can be used for dictionary induction, unsupervised machine translation, and transfer learning. In the unsupervised regime, an initial seed dictionary is learned in the absence of any known correspondences between words, through **distribution matching**, and the seed dictionary is then used to supervise the induction of the final alignment in what is typically referred to as a (possibly iterative) **refinement** step. We focus on the first step and compare distribution matching techniques in the context of language pairs for which mixed training stability and evaluation scores have been reported. We show that, surprisingly, when looking at this initial step in isolation, vanilla GANs are superior to more recent methods, both in terms of precision and robustness. The improvements reported by more recent methods thus stem from the refinement techniques, and we show that we can obtain state-of-the-art performance combining vanilla GANs with such refinement techniques.

## 1 Introduction

A word vector space – sometimes referred to as a *word embedding* – associates similar words in a vocabulary with similar vectors. Learning a projection of one word vector space into another, such that similar words – across the two word embeddings – are associated with similar vectors, is useful in many contexts, with the most prominent example being the alignment of vocabularies of different languages, i.e., word translation. This is a key step in machine translation of low-resource languages (Lample et al., 2018).

Projections between word vector spaces have typically been learned from seed dictionaries. In seminal papers (Mikolov et al., 2013; Faruqui and Dyer, 2014; Gouws and Søgaard, 2015), these seeds would comprise thousands of words, but Vulić and Korhonen (2016) showed that we can learn reliable projections from as little as 50 words. Smith et al. (2017) and Hauer et al. (2017) subsequently showed that the seed can be replaced with just words that are identical across languages; and Artetxe et al. (2017) showed that numerals can also do the job, in some cases; both proposals removing the need for an actual dictionary. Even more recently, entirely unsupervised approaches to projecting word vector spaces onto each other have been proposed, which induce seed dictionaries in the absence

| | INITIALIZATION AND OPTIMIZATION STEPS | | |
|---|---|---|---|
| **Authors** | **Unsupervised step** | **Supervised step** | **Extras** |
| Barone (2016) | GAN | None | |
| Zhang et al. (2017) | Wasserstein GAN | Procrustes | |
| Conneau et al. (2018) | GAN | Procrustes | |
| Hoshen and Wolf (2018) | ICP | Procrustes | Restarts |
| Alvarez-Melis and Jaakkola (2018) | Gromov-Wasserstein | Procrustes | |
| Artetxe et al. (2018) | Gromov-Wasserstein | Stochastic | |
| Yang et al. (2018) | Gromov-Wasserstein | MMD | |
| Xu et al. (2018) | GAN | Sinkhorn | Back-translation |
| Grave et al. (2018) | Gold-Rangarajan | Sinkhorn | |

Table 1: Approaches to unsupervised alignment of word vector spaces. We break down these approaches in two steps (and extras): (1) **Unsupervised** distribution matching for seed dictionary learning): (W)GANs, ICP, Gromov-Wasserstein initialization, and the convex relaxation proposed in Gold and Rangarajan (1996). (2) **Supervised** refinement: Procrustes, stochastic dictionary induction, maximum mean discrepancy (MMD), and the Sinkhorn algorithm.

of any known correspondences between words, using distribution matching techniques. These seed dictionaries are then used as supervision for alignment algorithms based on, e.g., Procrustes Analysis Schönemann (1966). These unsupervised systems, in other words, typically combine two steps: an unsupervised step of distribution matching and a (possibly iterative) (pseudo-)supervised step of refinement, based on a seed dictionary learned in the first step. See Table 1 for an overview.

The first unsupervised dictionary induction (UBDI) systems (Barone, 2016; Zhang et al., 2017; Conneau et al., 2018) were based on Generative Adversarial Networks (GANs) (Goodfellow et al., 2014). These approaches learn a linear transformation to minimize the divergence between a target distribution (say French word embeddings) and a source distribution (the English word embeddings projected into the French space). GAN-based approaches achieve impressive results for some language pairs (Conneau et al., 2018), but show instabilities for others. In particular, Søgaard et al. (2018) presented results suggesting that GAN-based UBDI is difficult for some language pairs exhibiting very different morphosyntactic properties, as well as when the monolingual corpora are very different. Recently, a range of unsupervised approaches that do not rely on GANs have been proposed (Artetxe et al., 2018; Hoshen and Wolf, 2018; Grave et al., 2018) in the hope they would provide a more robust alternative. In this paper, we show *none of these are more robust* on the language pairs we consider. Instead we propose a simple technique for making (vanilla) GAN-based UBDI more robust and show that combining this with a recently proposed refinement technique – stochastic dictionary induction (Artetxe et al., 2018) – leads to state-of-the-art performance in UBDI.

**Contributions** We present the first systematic comparison of (a subset of) recently proposed methods for UBDI. These methods are two-step pipelines of unsupervised distribution matching for seed induction and supervised refinement. While the authors typically introduce new approaches to both steps (see Table 1), distribution matching and refinement are independent, and in this paper, **we focus on the distribution matching step** - by either omitting refinement or using the same refinement method across different distribution matching, or seed dictionary induction methods. On the language pairs considered here, vanilla GANs are superior to more recently improved distribution matching techniques. Moreover, we show that using an unsupervised model selection method, we can often pick out the best vanilla GAN runs *in the absence of* cross-lingual supervision. Since vanilla GANs thus seem to remain an interesting technique for inducing seed dictionaries, we explore what causes the instability of vanilla GAN seed induction, by looking at how they perform on simple transformations of the embedding spaces, and by using a combination of supervised training and model interpolation to analyze the loss landscapes. The results lead us to conclude that the instability is caused by a mild form of mode collapse, that cannot easily be overcome by changes in the number of parameters, batch size, and learning rate. Nevertheless, vanilla GANs with unsupervised model selection seem superior to more recently proposed methods, and we show that when combined with a state-of-the-art refinement technique, vanilla GANs with unsupervised model selection is superior to these methods across the board.

## 2 GAN-initialized UBDI

In this section, we discuss the dynamics of GAN-based UBDI. While the idea of using GANs for UBDI originates with Barone (2016), we refer to Conneau et al. (2018) as the canonical implementation of GAN-based UBDI. Note that GANs are not a necessary component to unsupervised distribution matchning for alignment of vector spaces, albeit a popular approach (Barone, 2016; Conneau et al., 2018; Zhang et al., 2017). In §3, we briefly discuss how GAN-based initialization compares to the alternative of using point set registration techniques (Hoshen and Wolf, 2018) and related strategies.

A GAN consists of a generator and a discriminator (Goodfellow et al., 2014). The generator $G$ is trained to fool the discriminator $D$. The generator can be any differentiable function; in Conneau et al. (2018), it is a linear transform $\Omega$. Let $\mathbf{e} \in E$ be an English word vector, and $\mathbf{f} \in F$ a French word vector, both of dimensionality $d$. The goal of the generator is then to choose $\Omega \in \mathbb{R}^{d \times d}$ such that $\Omega E$ has a distribution close to $F$. The discriminator is a map $D_w : \mathcal{X} \to \{0, 1\}$, implemented in Conneau et al. (2018) as a multi-layered perceptron. The objective of the discriminator is to discriminate between vector spaces $F$ and $\Omega E$. During training, the model parameters $\Omega$ and $w$ are optimized using stochastic gradient descent by alternately updating the parameters of the discriminator based on the gradient of the discriminator loss and the parameters of the generator based on the gradient of the generator loss, which, by definition, is the inverse of the discriminator loss. The loss function used in Conneau et al. (2018) and in our experiments below is cross-entropy. In each iteration, we sample $N$ vectors $e \in E$ and $N$ vectors $f \in F$ and update the discriminator parameters $w$ according to $w \to w + \alpha \sum_{i=1}^{N} \nabla[\log D_w(f_i) + \log(1 - D_w(G_\Omega(e_i))]$.

Theoretically, the optimal parameters are a solution to the min-max problem: $\min_\Omega \max_w \mathbb{E}[\log(D_w(F)) + \log(1 - D_w(G_\Omega(E)))]$, which reduces to $\min_\Omega JS(P_F \mid P_\Omega)$. If a generator wins the game against an ideal discriminator on a very large number of samples, then $F$ and $\Omega E$ can be shown to be close in Jensen-Shannon divergence, and thus the model has learned the true data distribution. This result, referring to the distributions of the data, $p_{data}$, and the distribution, $p_g$, $G$ is sampling from, is from Goodfellow et al. (2014): If $G$ and $D$ have enough capacity, and at each step of training, the discriminator is allowed to reach its optimum given $G$, and $p_g$ is updated so as to improve the criterion $E_{\mathbf{x} \sim p_{data}}[\log D_G^*(\mathbf{x})]$ then $p_g$ converges to $p_{data}$. This result relies on a number of assumptions that do not hold in practice. The generator in Conneau et al. (2018), which learns a linear transform $\Omega$, has very limited capacity, for example, and we are updating $\Omega$ rather than $p_g$. In practice, therefore, during training, Conneau et al. (2018) alternate between $k$ steps of optimizing the discriminator and one step of optimizing the generator. Another common problem with training GANs is that the discriminator loss quickly drops to zero, when there is no overlap between $p_g$ and $p_{data}$ (Arjovsky et al., 2017); but note that in our case, the discriminator is initially presented with $IE$ and $F$, for which there is typically no trivial solution, since the embedding spaces are likely to overlap. We show in §4 that the discriminator and generator losses are poor model selection criteria, however; instead we propose a simple criterion based on cosine similarities between nearest neighbors in the learned alignment.

From $\Omega E$ and $F$, a seed (bilingual) dictionary can be extracted using nearest neighbor queries, i.e., by asking for the nearest neighbor of $\Omega E$ in $F$, or vice versa. Conneau et al. (2018) use a normalized nearest neighbor retrieval method to reduce the influence of hubs (Radovanović et al., 2010; Dinu et al., 2015). The method is called *cross-domain similarity local scaling* (CSLS) and used to expand high-density areas and condense low-density ones. The mean similarity of a source language embedding $\Omega \mathbf{e}$ to its $k$ nearest neighbors in the target language is defined as $\mu_E^k(\Omega(\mathbf{e})) = \frac{1}{k} \sum_{i=1}^{k} \cos(\mathbf{e}, \mathbf{f}_i)$, where $\cos$ is the cosine similarity. $\mu_F(\mathbf{f}_i)$ is defined in an analogous manner for every $i$. $CSLS(\mathbf{e}, \mathbf{f}_i)$ is then calculated as $2 \cos(\mathbf{e}, \mathbf{f}_i) - \mu_E(\Omega(\mathbf{e})) - \mu_F(\mathbf{f}_i)$. Conneau et al. (2018) use an unsupervised validation criterion based on CSLS. The translations of the top $k$ (10,000) most frequent words in the source language are obtained with CSLS and average pairwise cosine similarity is computed over them. This metric is considered indicative of the closeness between the projected source space and the target space, and is found to correlate well with supervised evaluation metrics. After inducing a bilingual dictionary, $E_d$ and $F_d$, by querying $\Omega E$ and $F$ with CSLS, Conneau et al. (2018) perform a refinement step based on Procrustes Analysis (Schönemann, 1966). Here, the optimal mapping $\Omega$ that maps the words in the seed dictionary onto each other, is computed analytically as $\Omega = UV^T$, where $U$ and $V$ are obtained via the singular value decomposition $U\Sigma V^T$ of $F_d^T E_d$.

# 3   Alternatives to GAN-initialized UBDI

This section introduces some recent alternatives to (vanilla) GAN-initialized UBDI. In Table 1, we list more approaches and classify them by how they perform unsupervised distribution matching and supervised refinement.

**Iterative closest point**   The idea of minimizing nearest neighbor distances for unsupervised model selection is also found in point set registration and lies at the core of iterative closest point (ICP) optimization (Besl and McKay, 1992). ICP typically minimizes the $\lambda_2$ distance (mean squared error) between nearest neighbor pairs. The ICP optimization algorithm works by assigning each transformed vector to its nearest neighbor and then computing the new relative transformation that minimizes the cost function with respect to this assignment. ICP can be shown to converge to local optima (Besl and McKay, 1992), in polynomial time (Ezra et al., 2006). ICP easily gets trapped in local optima, however, exact algorithms only exist for two- and three-dimensional point set registration, and these algorithms are slow (Yang et al., 2016). Generally, it holds that the optimal solution to the GAN min-max problem is also optimal for ICP. To see this, note that a GAN minimizes the Jensen-Shannon divergence between $F$ and $\Omega E$. The optimal solution to this is $F = \Omega E$. As sample size goes to infinity, this means the $\mathcal{L}_2$ loss in ICP goes to 0. In other words, the ICP loss is minimal if an optimal solution to the UBDI min-max problem is found. ICP was independently proposed for UBDI in Hoshen and Wolf (2018). They report their method only works using PCA initialization, i.e. they project a subset of both sets of word embeddings onto the 50 first principal components, and learn an initial seed dictionary using ICP on the lower-dimensional embeddings. This seed mapping is then used as starting point for ICP on the full word embeddings. We explored PCA initialization for GAN-based distribution matching, but observed the opposite effect, namely that PCA initialization leads to a degradation in performance. The most important thing to note from Hoshen and Wolf (2018), however, is that they do 500 random restarts of the PCA initialization to obtain robust performance; ICP, in other words, is extremely sensitive to initialization. This explains their poor performance under our experimental protocol below (Table 2).

**Wasserstein GAN**   Zhang et al. (2017) were the first to introduce Wasserstein GANs as a way to learn seed dictionaries in the context of UBDI. In their best system, they train simple Wasserstein GANs and use the resulting seed dictionaries to supervise Procrustes Analysis. We modified the MUSE code to experiment with Wasserstein GANs in a controlled way. Simple Wasserstein GANs were unsuccessful, but with gradient penalty (Gulrajani et al., 2017), we obtained almost competitive results, after tuning the learning rate and the gradient penalty $\lambda$ using nearest neighbor cosine distance as validation criterion. On the other hand, the results were not significantly better, and instability did not improve. Finally, we experimented with CT-GANs (Wei et al., 2018), an extension of Wasserstein GANs with gradient penalty, but this only lowered performance and increased instability. Since Wasserstein GANs and CT-GANs were consistently worse and less stable than vanilla GANs, we do not include them in the experiments below.

**Gromov-Wasserstein**   Alvarez-Melis and Jaakkola (2018) present a very different initialization strategy. In brief, Alvarez-Melis and Jaakkola (2018) learn a linear transformation to minimize Gromov-Wasserstein distances of distances between nearest neighbors, in the absence of cross-lingual supervision. We report the performance of their system in the experiments below, but results (Table 2) were all negative. We think the reason is that Alvarez-Melis and Jaakkola (2018) only consider small subsamples of the vector spaces, and that in hard cases, alignments induced on subspaces are unlikely to scale. It achieved an impressive P@1 of 85.6 on the Greek MUSE dataset (Conneau et al. (2018) obtain 59.5); but on the datasets, where Conneau et al. (2018) are instable, considered here, it consistently fails to align the vector spaces.

Artetxe et al. (2018) introduce a very simple, related initialization method that is also based on Gromov-Wasserstein distances of distances between nearest neighbors: They use these second-order distances to build a seed dictionary directly by aligning nearest neighbors across languages. By itself, this is a poor initialization method (see Table 2). Artetxe et al. (2018), however, combine this with a new refinement method called *stochastic dictionary induction*, i.e., randomly dropping out dimensions of the similarity matrix when extracting a seed dictionary for the next iteration of Procrustes Analysis. Artetxe et al. (2018) show in an ablation study for one language pair (English-Finnish) that the initialization method only works in combination with the stochastic dictionary induction step, i.e.,

| | | TO ENGLISH | | | | | | | | | | | | |
|---|---|---|---|---|---|---|---|---|---|---|---|---|---|---|
| | | et | | fa | | fi | | lv | | tr | | vi | | **av** | |
| | | max | fail | max | fail | max | fail | max | fail | max | fail | max | fail | max | fail |
| NO REFINEMENT | | | | | | | | | | | | | | | |
| Conneau et al. (2018) | GAN | **6.4** | 9 | **22.5** | 3 | **28.5** | 1 | **14.3** | 9 | **32.1** | 2 | **2.4** | 9 | **17.7** | 5.5 |
| Hoshen and Wolf (2018) | ICP | 0.1 | 10 | 0 | 10 | 0 | 10 | 0 | 10 | 0 | 10 | 0 | 10 | 0 | 10 |
| Artetxe et al. (2018) | GW | 0 | 10 | 0.1 | 10 | 0.1 | 10 | 0.1 | 10 | 0.1 | 10 | 0.1 | 10 | 0.1 | 10 |
| Alvarez-Melis and Jaakkola (2018) | GW | 0 | 10 | 0 | 10 | 0 | 10 | 0 | 10 | 0 | 10 | 0 | 10 | 0 | 10 |
| WITH PROCRUSTES REFINEMENT | | | | | | | | | | | | | | | |
| Conneau et al. (2018) | GAN | **27.5** | 9 | **40.9** | 3 | 58.9 | 1 | **33.2** | 9 | **60.6** | 2 | **51.3** | 9 | **45.4** | 5.5 |
| Hoshen and Wolf (2018) | ICP | 0.1 | 10 | 0 | 10 | 0 | 10 | 0 | 10 | 0 | 10 | 0 | 10 | 0 | 10 |
| Artetxe et al. (2018) | GW | 1.1 | 10 | 40.2 | 0 | **60.5** | 0 | 0.1 | 10 | 59.6 | 0 | 0.3 | 10 | 27.0 | 5 |
| Alvarez-Melis and Jaakkola (2018) | GW | 0 | 10 | 0 | 10 | 0 | 10 | 0 | 10 | 0 | 10 | 0 | 10 | 0 | 10 |

Table 2: Comparisons of unsupervised **seed dictionary** learning strategies *in the absence of refinement* (upper half) or *using the same refinement technique* (orthogonal Procrustes) (lower half). For results with refinement, we use GANs, ICPs, and Gromov-Wasserstein (GW) distribution matching and feed seed dictionaries to Procrustes refinement. We then report maximum performance (P@1) and stability (fails) across 10 runs. We consider a P@1 score below 2% a failure. The results suggest that GANs, in spite of their instability, have the highest potential for inducing useful seed dictionaries.

without the application of stochasticity, the induced mapping is degenerate. In our experiments below, we show that this finding generalizes to other language pairs, suggesting that the stochastic dictionary induction is the main contribution in their work. We show that when combined with vanilla GANs for the initial step of learning a seed dictionary through distribution matching, stochastic dictionary induction performs even better.

**Convex Relaxation** The Gold-Rangarajan relaxation is a convex relaxation of the (NP-hard) graph matching problem and can be solved using the Frank-Wolfe algorithm. Once the minimal optimizer is computed, an initial transformation is obtained using singular-value decomposition. The Gold-Rangarajan relaxation can thus be used for stable learning of seed dictionaries Grave et al. (2018). It remains an open question how this strategy fairs on challenging language pairs such as the ones included here. We would have liked to include this approach in our experiments, but the code was not publicly available at the time of writing.

**Properties of Unsupervised Alignment Algorithms** The above approaches provably work if the two vector spaces to be aligned, are isomorphic, except for the pathological case where the vectors are placed on an equidistant grid forming a sphere.[1] A function $\Omega$ from $E$ to $F$ is a linear transformation if $\Omega(f + g) = \Omega(f) + \Omega(g)$ and $\Omega(kf) = k\Omega(f)$ for all elements $f, g$ of $E$, and for all scalars $k$. An invertible linear transformation is called an *isomorphism*. The two vector spaces $E$ and $F$ are called isomorphic, if there is an isomorphism from $E$ to $F$. Equivalently, if the kernel of a linear transformation between two vector spaces of the same dimensionality contains only the zero vector, it is invertible and hence an isomorphism. Most work on supervised or unsupervised alignment of word vector spaces relies on the assumption that they are approximately isomorphic, i.e., isomorphic after removing a small set of vertices (Mikolov et al., 2013; Barone, 2016; Zhang et al., 2017; Conneau et al., 2018). It is not difficult to show that many pairs of vector spaces are not approximately isomorphic, however. See Søgaard et al. (2018) for examples.

# 4 Experiments

In our experiments, we focus on aligning word vector spaces between two languages, by projecting from the foreign language into English. Our languages are: Estonian (et), Farsi (fa), Finnish (fi), Latvian (lv), Turkish (tr), and Vietnamese (vi). This selection of languages is motivated by observed instability when training vanilla GANs, e.g., Søgaard et al. (2018). In addition, the languages span four language families: Finno-Ugric (et, fi), Indo-European (fa, lv), Turkic (tr), and Austroasiatic (vi).

**Data** In all our experiments, we use pretrained FastText embeddings (Bojanowski et al., 2017) and the bilingual test dictionaries released along with the MUSE system.[2] The FastText embeddings are trained on Wikipedia dumps[3]; the bilingual dictionaries were created using an in-house Facebook translation tool and contain translations for 1500 test words for each language pair. Since we cannot do reliable hyper-parameter optimization in the absence of cross-lingual supervision, we use MUSE with the default parameters (Conneau et al., 2018). For the experiments with stochastic dictionary induction (Table 3), we use the implementation in the VecMap framework (Artetxe et al., 2018).[4]

## 4.1 Comparison of distribution matching strategies

Our main experiments, reported in Table 2, compare the initialization strategies listed in Table 2: vanilla GANs, the two varieties of Gromov-Wasserstein (see §3), and ICP.[5] Table 2 is split in two: First we report the performance, measured as precision at one, in the absence of refinement; and then we report the performance *with* refinement, using *the same* refinement technique (Procrustes Analysis) across the board. For all the randomly initialized algorithms (the first three), we report the best of 10 runs and the number of *fails*, where fails are runs with scores lower than 2%.[6] The reported scores are P@1, i.e., the fraction of words whose neighbors are translation equivalents.

We believe it is crucial to evaluate the different techniques this way, instead of simply comparing the numbers reported in the relevant papers: First of all, no three of these authors report performance on the same datasets. Secondly, if the authors use different refinement techniques, it is impossible to see the impact of the initialization strategies in the reported numbers. Instead we control for the refinement techniques and study the distribution matching techniques in Table 1 in isolation. This means, for example, that we evaluate the Artetxe et al. (2018) in the absence of stochastic dictionary induction, and Hoshen and Wolf (2018) in the absence of 500 random restarts. In §4.2 (Table 3), we compare vanilla GANs and Gromov-Wasserstein in the context of stocastic dictionary induction.

The patterns in Table 2 are very consistent. Vanilla GAN distribution matching is very instable, with 1/10 fails for Finnish and Turkish, but 6, 7 and 9 fails for Estonian, Latvian, and Vietnamese, respectively. All other methods are *more* instable, however, with the distribution matching techniques in Hoshen and Wolf (2018) and Alvarez-Melis and Jaakkola (2018) failing across the board, with or without supervised Procrustes refinement. Vanilla GAN distribution matching also leads to higher precision for 5/6 language pairs.

Vanilla GAN distribution matching thus seems to have the highest potential for inducing useful seed dictionaries among all these methods. If we could only manage their instability, GANs seem to provide us with a better point of departure. This naturally leads us to ask: *Is it feasible to select good vanilla GAN UBDI runs from a batch of random restarts, in the absence of cross-lingual supervision?* This question is explored in §4.2, in which we also explore whether state-of-the-art performance can be achieved with vanilla GANs and a more advanced refinement technique, namely stochastic dictionary induction.

|  | PROCRUSTES | STOCHASTIC DICTIONARY INDUCTION | |
|---|---|---|---|
|  | C-MUSE | C-MUSE | Artetxe et al. (2018) |
| et-en | 27.5 | 47.6 | 47.6 |
| fa-en | 40.9 | **41.5** | 40.2 |
| fi-en | 58.9 | 62.5 | **63.6** |
| lv-en | 33.2 | **44.1** | 41.6 |
| tr-en | 60.6 | **62.8** | 60.6 |
| vi-en | 51.3 | **54.3** | 0.3 |
| **average** | 45.4 | **52.1** | 42.3 |

Table 3: Comparison of MUSE with cosine-based model selection over 10 random restarts (C-MUSE) with and without stochastic dictionary induction (with suggested hyper-parameters from Artetxe et al. (2018)), against state of the art. Using vanilla GANs is better than Gromov-Wasserstein on average and better on 4/6 language pairs.

## 4.2 GAN distribution matching with random restarts

Exploring this question we found that the discriminator loss during training, which is used as a model selection criterion in Daskalakis et al. (2018), is a poor selection criterion. However, we did find another unsupervised model selection criterion that correlates well with UBDI performance: cosine similarity of (induced) nearest neighbors. This criterion is also used as a stopping criterion in Conneau et al. (2018), and can be used with or without CSLS scaling. This stopping criterion in fact turns out to be a quite robust model selection criterion for picking the best out of $n$ random restarts.

In Table 3, we compare MUSE with 10 random restarts and using CSLS cosine similarity of nearest neighbors as an unsupervised model selection criterion, to the full state-of-the-art model in Artetxe et al. (2018) *with* stochastic dictionary induction. What we see in these results, is that Artetxe et al. (2018) is still superior to MUSE with random restarts, but even with 10 restarts, the gap narrows considerably, and MUSE is better on 2/6 languages. Note, however, that this is a comparison of two systems using two different refinement techniques. If we combine vanilla GAN distribution matching from MUSE with the stochastic dictionary induction technique from Artetxe et al. (2018), we obtain slightly better performance than Artetxe et al. (2018) (Table 3, mid-column): While overall improvements are small, compared to the differences in seed dictionary quality, the combination of vanilla GANs for distribution matching and stochastic dictionary induction provides a promising and fully competitive alternative to the state of the art for unsupervised word translation.

## 4.3 Discussion and Further Experiments

We have shown that while vanilla GANs are instable, they carry a seemingly unique potential for UBDI. We have shown that a simple unsupervised cosine-based model selection criterion can achieve robust state-of-the-art performance. We have performed several other experiments to probe this instability in search of ways to stabilize vanilla GANs without significant performance drops. This subsection summarizes these experiments.

**Normalization** We observed that GAN-based UBDI becomes more instable and performance deteriorates with unit length normalization. We performed unit length normalization (ULN) of all vectors $\mathbf{x}$, i.e., $\mathbf{x}' = \frac{\mathbf{x}}{||\mathbf{x}||^2}$, which is often used in supervised bilingual dictionary induction (Xing et al., 2015; Artetxe et al., 2017). We used this transform to project word vectors onto a sphere – to control for shape information. If vectors are distributed smoothly over two spheres, there is no way to learn an alignment in the absence of dictionary seed; in other words, if vanilla GAN distribution matching is unaffected by this transform, vanilla GANs learn from density information alone. While supervised methods are insensitive to or benefit from ULN, we find that vanilla GANs are very sensitive to such normalization; in fact, the number of failed runs over six languages increases from below 50% to 90%. For example, while for Finnish, MUSE only fails in 1/10 runs, MUSE with ULN failed across the board; for Farsi, MUSE with ULN failed in 6/10 runs, compared to 3/10. We verify that supervised alignment is not affected by ULN by running Procrustes refinement with a seed dictionary as supervision; here, performance remains unchanged under this transformation.

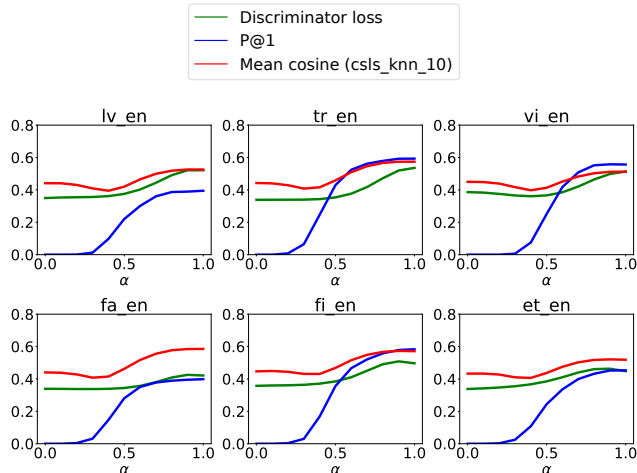

Figure 1: Discriminator loss averaged over all training data points (green), P@1 on the test data points (blue) and mean cosine similarity (red) on the training data – for generator parameters on the line segment that connects the unsupervised GAN solution with the supervised Procrustes Analysis solution. $\alpha$ is the interpolation parameter moving the generator parameters from the unsupervised GAN solution ($\alpha = 0$) to the supervised solution ($\alpha = 1$).

**Noise injection**   On the contrary, GAN-based UBDI is largely unaffected by noise injection. We saw this from running experiments on a few languages, but do not report performance across the board. Specifically, we add 25% random vectors, randomly sampled from a hypercube bounding the vector set. GAN-based UBDI results are not affected by noise injection. This, we found, is because the injected vectors rarely end up in the seed dictionaries used for subsequent refinement.

**Over-parameterization**   GAN training is instable because discriminators end up in poor local optima or saddle points (see below). A known technique for escaping local optima is over-parameterization (Brutzkus et al., 2018). We experimented with widening our discriminators to smoothen the loss landscape. Results were mixed, with more stability and better performance on some languages, and less stability and worse performance on others. We provide the full list of results in the Appendix.

**Large batches and small learning rates**   Previous work has shown that large learning rate and small batch size contribute towards SGD finding flatter minima (Jastrzebski et al., 2018), but in our experiments, we are interested in the discriminator not ending up in flat regions, where there is no signal to update the generator. We therefore experiment with (higher and) *smaller* learning rate and (smaller and) *larger* batch sizes. The motivation behind both is decreasing the scale of random fluctuations in the SGD dynamics (Smith and Le, 2017; Balles et al., 2017), enabling the discriminator to explore narrower regions in the loss landscape. Increasing the batch size or varying the learning rate (up or down), however, leads to worse performance, and it seems the MUSE default hyperparameters are close to optimal. We provide the full list of results in the Appendix.

**Exploring the loss landscapes**   GAN training instability arises from discriminators getting stuck in saddle points, where neither the discriminator nor the generator has a learning signals. To show this, we analyze the discriminator loss in areas of convergence by plotting it as a function of the generator parameters. Specifically, we plot the loss surface along its intersection with a line segment connecting two sets of parameters (Goodfellow et al., 2015; Li et al., 2018). In our case, we interpolate between the model induced by GAN-based UBDI and the (oracle) model obtained using supervised Procrustes Analysis. Results are shown in Figure 1. The green loss curves represent the current discriminator's loss along all the generators between the current generator and the generator found by Procrustes refinement. We see that while performance (P@1 and mean cosine similarity) goes up as soon as we move closer toward the supervised solution, the discriminator loss does not change until we get very close to this solution, suggesting there is no learning signal in this direction for GAN-based UBDI.

This is along a line segment representing the shortest path from the failed generator to the oracle generator, of course; linear interpolation provides no guarantee there are no almost-as-short paths with plenty of signal. A more sophisticated sampling method is to sample along two random direction vectors (Goodfellow et al., 2015; Li et al., 2018). We used an alternative strategy of sampling from normal distributions with fixed variance that were orthogonal to the line segment. We observed the same pattern, leading us to the conclusion that instability is caused by discriminator saddle points.

## 5 Conclusions

This paper explores the dynamics of (vanilla) GAN training in the context of unsupervised word translation and a systematic comparison of GANs with different distribution matching (seed induction) methods across six challenging language pairs. Our main finding is that vanilla GANs, in spite of their instability, have the highest potential for inducing useful seed dictionaries. We explore an unsupervised model selection criterion for selecting the best models from multiple random restarts, narrowing the gap between MUSE and Artetxe et al. (2018), and further show that combining GANs with stochastic dictionary induction provides a new state of the art for unsupervised word translation.

## Acknowledgements

We thank the anonymous reviewers for their comments and suggestions. Mareike Hartmann was supported by the Carlsberg Foundation. Anders Søgaard was supported by a Google Focused Research Award.

## Footnotes

[1] In this case, there is an infinite set of equally good linear transformations (rotations) that achieve the same training loss. Similarly, for two binary-valued, $n$-dimensional vector spaces with one vector in each possible position. Here the number of local optima would be $2^n$, but since the loss is the same in each of them the loss landscape is highly non-convex, and the basin of convergence is therefore very small (Yang et al., 2016). The chance of aligning the two spaces using gradient descent optimization would be $\frac{1}{2^n}$. In other words, minimizing the Jensen-Shannon divergence between the word vector distributions, even in the easy case, is not always guaranteed to uncover an alignment between translation equivalents. From the above, it follows that alignments between linearly alignable vector spaces cannot always be learned using UBDI methods. In §3.1 , we test for approximate isomorphism to decide whether two vector spaces are linearly alignable.§3.2–3.3 are devoted to analyzing *when* alignments between linearly alignable vector spaces can be learned.

[2]https://github.com/facebookresearch/MUSE

[3]https://fasttext.cc/docs/en/pretrained-vectors.html

[4]https://github.com/artetxem/vecmap

[5]We ignore Wasserstein GANs, which proved more instable than vanilla GANs in our preliminary experiments, as well as Gold-Rangarajan, which performs considerably below current state of the art.

[6]In practice, performance tends to be much higher than 2% for successful runs, hence slight changes in the threshold value would not affect results.

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
