[Supplementary Material · appendix.pdf]

# Appendix A

| | to English | | | | | | | | | | | | | |
| --- | --- | --- | --- | --- | --- | --- | --- | --- | --- | --- | --- | --- | --- | --- |
| | et | | fi | | fa | | lv | | tr | | vi | | **av** | |
| | max | fail | max | fail | max | fail | max | fail | max | fail | max | fail | max | fail |
| *Unit Length (UL) Normalization* | | | | | | | | | | | | | | |
| Default: Centering | 27.5 | 4 | 58.9 | 1 | 40.5 | 2 | 33.2 | 4 | 60.1 | 0 | 0 | 5 | 36.7 | 2.7 |
| Centering + UL Normalization | 0 | 5 | 0 | 5 | 39.7 | 4 | 0 | 5 | 0 | 5 | 0 | 5 | 6.6 | 4.8 |
| *Noise* | | | | | | | | | | | | | | |
| Default: No Noise | 27.5 | 4 | 58.9 | 1 | 40.5 | 2 | 33.2 | 4 | 60.1 | 0 | 0 | 5 | 36.7 | 2.7 |
| with Noise | 31.5 | 4 | 57.8 | 1 | 39.9 | 1 | 0.5 | 4 | 59.8 | 1 | 0 | 5 | 31.6 | 2.7 |
| *Batch size* | | | | | | | | | | | | | | |
| default: bs=32 | 27.5 | 4 | 58.9 | 1 | 40.5 | 2 | 33.2 | 4 | 60.1 | 0 | 0 | 5 | 36.7 | 2.7 |
| bs=16 | 25.6 | 3 | 58.1 | 0 | 40.9 | 2 | 0 | 5 | 59.9 | 1 | 53.4 | 4 | 39.7 | 2.5 |
| bs=64 | 0 | 5 | 58.3 | 1 | 39.5 | 3 | 0 | 5 | 59.1 | 1 | 0 | 5 | 26.2 | 3.3 |
| *Hidden dim* | | | | | | | | | | | | | | |
| default: hd=2048 | 27.5 | 4 | 58.9 | 1 | 40.5 | 2 | 33.2 | 4 | 60.1 | 0 | 0 | 5 | 36.7 | 2.7 |
| hd=4096 | 0 | 5 | 58.0 | 0 | 39.9 | 0 | 0 | 5 | 60.2 | 1 | 0.2 | 4 | 26.4 | 2.5 |

Table 1: Results for experiments with normalization, noise injection, variation in batch size (BS) and dimension of the hidden layer in the discriminator (HD). All results per language are averaged over 5 runs. The last column shows the average over all languages the experiment was conducted for.

| | from English | | | |
| --- | --- | --- | --- | --- |
| | el | | hu | |
| | max | fail | max | fail |
| *Default* | | | | |
| bs=32, hd=2048, lr=0.1 | 44.9 | 1 | 42.8 | 1 |
| *Batch Size* | | | | |
| bs=16 | 36.6 | 2 | 37.4 | 0 |
| bs=128 | 0 | 5 | 9.4 | 4 |
| bs=1600 | 0 | 5 | 0 | 5 |
| bs=16000 | 0 | 5 | 0 | 5 |
| *Learning Rate* | | | | |
| lr=0.01 | 29.45 | 1 | 43.4 | 1 |
| lr=0.001 | 0 | 5 | 0 | 5 |
| *Hidden dim* | | | | |
| hd=8192 | 28.6 | 1 | 45.5 | 0 |

Table 2: English to Greek (EL) and English to Hungarian (HU) results for experiments with variation in batch size (BS), dimension of the hidden layer in the discriminator (HD) and learning rate (LR). All results per language are averaged over 5 runs. Embeddings were centered and UL normalized.