[Reviews · NeurIPS 2019]

Reviewer 1



Although this paper only combines existing techniques, it's good science to do more careful experimentation on existing techniques and new combinations of existing techniques. I think this is a valuable contribution. The heart of the paper is in sections 4.1 and 4.2. I could be mistaken, but the impression I get is that these sections read like two different projects combined together. Section 4.1 shows clearly and convincingly that more recent alternatives to GAN distribution matching don't do as well on more difficult language pairs. Some small things that could strengthen this section are: - How was the 2% threshold for defining "failure" chosen? Could you use some other metric that doesn't require you to choose an arbitrary threshold? And reporting the max over 10 runs might be misleading since the variance is so high. Would it be better to report mean and standard deviation, or even plot all 10 scores? - Restate results on less difficult language pairs too, to provide a complete picture, so that the takeaway message is something like "for languages with properties ABC, use method D, but for languages with properties EFG, use method H." - The paper presenting Gromov-Wasserstein alignment points out that their method is much faster than GANs. This would be worth mentioning here too. I found Section 4.2 difficult to understand. It took me a couple of reads to even realize that this section contained both of the main innovations of the paper. I would have expected this section to be written as follows: - Subsection 4.2.1: GAN with Procrustes (Table 2, bottom half) is compared with GAN with SBDI, and is found to have a higher average or maximum score over 10 runs. - Subsection 4.2.2: Using 10 runs of GAN+SBDI, model selection using discriminator loss is compared against model selection using cosine similarity, and the latter is found to have a higher score. (The opposite order would be possible too: show that cosine-similarity model selection works well on GAN+Procrustes, then show that GAN+SBDI is better.) But neither of these comparisons is made. First, it's stated that model selection using discriminator loss doesn't work (line 241); as far as I can tell, no justification is made for this claim. Second, a comparison is made between C-MUSE+Procrustes vs G-W+SBDI. The authors admit this is not an apples-to-apples comparison, so they make another comparison between C-MUSE+Procrustes and C-MUSE+SBDI. But this comparison fails to make a connection with Section 4.1 in two ways. - I assume that what is called GAN in Section 4.1 is the same as what is called MUSE in Section 4.2. This is not stated anywhere, as far as I can tell, and if I'm mistaken, then I don't know how to relate 4.1 and 4.2 at all. - Section 4.2 adds cosine-similarity model selection without comparing it to any results in Section 4.1.

Reviewer 2



This paper is a good survey paper that introduces unsupervised word translation methods, especially GAN based methods. However, this is not a paper that proposes a new method of technique. This could be a good workshop paper related with unsupervised translation lexicon acquisition, but is not enough for a full NeurIPS paper.

Reviewer 3



Originality: This paper is mostly about empirical analysis. Although there's not much originality in algorithmic proposals, it provides a critical view of the current methods. After all they also achieve SoTA on some alignment language pairs. Quality: This paper looks at the current methods for embedding alignment with a critical eye -- to see what really works best using the same pairs of languages. The summary of related work is quite illuminating. Clarity: The paper is very well written and very clear, at least for someone who are somewhat familiar with embeddings and word alignment. Significance: I think researchers and practitioners will find this useful.

[Author Response · NeurIPS 2019]



1 Thanks to all of you for your thoughtful reviews and very useful suggestions.

## Reviewer 1

a) "it's good science to do more careful experimentation on existing techniques and new combinations of existing techniques. I think this is a valuable contribution." **RESPONSE:** We agree and believe systematic comparisons of unsupervised cross-lingual learning methods are particularly important at a time where this area is getting very crowded.

b) "How was the 2% threshold for defining "failure" chosen?" **RESPONSE:** We follow previous work in using an absolute threshold, as well as maximum scores. (Artetxe et al., 2018, uses 5%, for example.) In practice, performance for unsuccessful runs tends to be either >.1 or 0, so a different threshold would be unlikely to change results. We will include mean and standard deviation in the revised version of the paper, but note that maxima highlight the potential of methods.

c) We are also happy to include results on less difficult language pairs, but would like to point out that unsupervised cross-lingual learning is *only relevant for low-resource languages*, which tend to be typologically different from English/Spanish and therefore difficult.

d) "I found Section 4.2 difficult to understand." **RESPONSE:** Thanks for the suggestions, which we will implement in the revised version.

e) "But this comparison fails to make a connection with Section 4.1 in two ways." **RESPONSE:** Sorry if this was not clear: MUSE is the FAIR system consisting of GAN+Procrustes, so GAN=C-MUSE. Both C-MUSE+Procrustes and C-MUSE+SBDI use cosine-based model selection (csls).

## Reviewer 2

a) We agree our paper presents a "detailed and fair comparison" and "show that combining GANs with stochastic dictionary induction gives a new state of the art". We do not agree this "is not enough for a full NeurIPS paper." This is a crowded area, with new methods being proposed all the time. The world does not necessarily need more methods, but to understand what works (when), and what does not.

b) You state that our paper needs "a new insight or method that improves the current performances of unsupervised word translation methods". While this was not our main goal, we do, as you say, "show that combining GANs with stochastic dictionary induction gives a new state of the art". This, we believe, qualifies as an insight improving the current performance of our methods.

## Reviewer 3

a) We agree our main contribution is "to fairly compare many methods in a standardized fashion", and that, in addition, we also present a new model selection criterion and establish a new state of the art.

b) We like the idea of "aligning 3 or more languages in a shared embedding space", but this goes well beyond the standard scenario explored in this paper.



[Meta-Review · NeurIPS 2019]

Accept. The evaluation performed in the paper can be useful to people working in this field.